# Roll Motion of a Water Filled Floating Cylinder—Additional Experimental Verification

**Roman Gabl** [1,2,*] , **Thomas Davey** [1] **and David M. Ingram** [1]

[1] School of Engineering, Institute for Energy Systems, FloWave Ocean Energy Research Facility, The University of Edinburgh, Max Born Crescent, Edinburgh EH9 3BF, UK; tom.davey@flowave.ed.ac.uk (T.D.); David.Ingram@ed.ac.uk (D.M.I.)

[2] Unit of Hydraulic Engineering, University of Innsbruck, Technikerstraße 13, 6020 Innsbruck, Austria

[*] Correspondence: roman.gabl@ed.ac.uk

**Abstract:** Understanding the behaviour of water filled bodies is important from an applied engineering perspective when understanding the sea-keeping performance of certain floating platforms and vessels. Even by assuming that the deformation is negligible small in relation to the motion of the structure, these fluid-structure-fluid interactions are challenging to model, both physically and numerically, and there is a notable lack of reference data sets and studies to support the validation of this work. Most of the existing information is highly specific to certain hulls forms, or is limited to small motions. A previous study addressed this by modelling a floating cylinder (giving a more generic case) with roll and pitch motions in excess of 20°. The presented experiment expands on that work to further investigate the previously observed switch between pitch and roll in the cylinder under wave action as induced by the sloshing of the internal water volume. An additional experimental investigation, focused on a single draft, was conducted to test open research questions from the previous study. Here we show that the roll response of the water filled cylinder is repeatable, independent of the tank position and wave amplitude, provided the observation time is long enough to capture the fully developed motion response of the floating object. The mooring system used comprised four soft lines connected on two points on the cylinder. This arrangement resulted in slightly different restoring forces in different wave directions. A relative change of the wave direction by 90° led to a larger wave frequency band in which the roll motion occurred. These cases were, again, also conducted with the solid ballast. Both sets of data provide an interesting validation case for future work on water ballast inside a floating object.

**Keywords:** floating cylinder; water filled; motion capturing; wave tank; wave gauges; free surface; sloshing; validation experiment

---

## 1. Introduction

Fluid-Structure-Fluid-Interactions (i.e., the behaviour of fluids within a floating body) influence the behaviour of a number of floating platforms, vessels and devices (e.g., certain wave energy converters) under the assumption of a negligible small deformation in relation to the large motions. Motivated by the paucity of studies and data relating to generic vessel shapes with large motions, which in turn limited numerical model validation, a previous study by Gabl et al. [1] examined a water filled floating cylinder. This experimental investigation explored four different drafts and two different ballast options, namely water and solid, under regular waves with a variable wave frequency, with the full dataset available for modelling validation purposes [2]. Large motions in the range of up to 20° were observed and allowed the investigation of the influence of the sloshing of the inside water on the motion of the structure. It could be shown that the water filled cylinder showed a switch from pitch to

roll (rotation around the axis in the wave direction) for a particular frequency band. This could not be observed in the solid ballast option and is consequently concluded to be caused, or at least enhanced, by the motion of the inner water body. The work presented here extends the work from the previous investigation, but with a narrower scope. The open research questions (RQ) focus on the position of the apparatus in the test tank, the importance of the observation time, sensitivity to wave amplitude, and sensitivity to wave direction, as well as the influence of the mooring system. These RQ are further discussed in Section 3.1. The observations from the previous experimental programme are used to narrow down the experimental scope in terms of cylinder draft and ballast variables, allowing this study to concentrate to the interesting behaviours observed in pitch/roll response switching and its connection to outlined research questions.

Fossen and Nijmeimer [3] define parametric resonance as a system under external excitation including a time-varying parameter. This can lead to oscillations larger than caused by resonance, under which the acting force varies at the same frequency as the natural frequency of the system. Such a behaviour can be observed in ships, for which it is called parametric rolling. It can also occur for ships sailing against waves with a length close to the total length of the vessel, which can lead to a wave excitation frequency near twice the natural roll frequency. A change of the direction as well as the speed can reduce this effect significantly as well as adequate roll damping. A number of particularly severe incidents involving container ships and roll motions of up to 40° intensified the research efforts in this particular field [3,4], but smaller roll angles will also have a significant influence on vessel usability and their detailed design [5,6]. A wide range of numerical models are suppose to predict the roll motion correctly [7–10] along with experimental investigations which provide case-specific validation data [11,12]. Zhou et al. [13] and Zhou [14] showed good agreement between experimental investigations and a hybrid model, which connects a 3D-numerical calculation of the roll damping coefficient and in a second step the solution of the 3 Degree of Freedom (DoF) motion of the ship. Specific investigations comparing liquid and frozen ballast showed that there is no significant influence of the roll response [15,16].

The cylindrical geometry used in this investigation is most akin to a cylindrical platform [17,18], but it not intended to model any specific prototype. It is instead intended to provide a generic geometry which removes the influence of specific hull forms and ensures the data is more usable in numerical modelling development and validation (at which point hull form influences can be reintroduced). The hull form of ships has a significant influence on the behaviour but it is far more complex than the presented geometry and often impractical for early stage modelling implementation. Tarrant and Meskell [19] investigated parametric rolling for point absorbers deployed as wave energy converter (WEC) with two relative oscillating bodies (Wavebob). They describe this phenomena as a coupling of two DoF when wave frequency is twice of the natural frequency. The fluid structure interaction in the time domain was supported by WAMIT calculations and validated with a 1:17 scaled model. Radhakrishnan et al. [20] presented a study of a spherical buoy moored with a single line in three different depth. The sway motion caused by the instability was depending on the relative position to the still water position and could be observed in the frequency range of 1.5 to 2.25 with a peak around double the natural period.

As observed in the previous floating cylinder, this roll motion only occurs with the water ballast option [1] (i.e., not when a solid internal ballast was utilised). Hence, it can be assumed that this instability is caused, or at least intensified, by the sloshing of the inner water body. Sloshing effects are widely studied using numerical and experimental approaches and have many different applications. For example, Jiang et al. [21] studies the free surface inside a rectangular tank introduced by a big jet. More common is the usage of an oscillation machine [22] or 6-DoF sloshing platforms [23]. Those experimental investigations are used as a validation experiment for numerical simulation and, in particular, the meshless Smoothed Particle Hydrodynamics (SPH) method which has the potential to combine big motions of floating structures with sloshing [24–29]. The geometries include cylindrical and rectangular forms [30] but also more complex geometries such as fuel tanks [31] and other large

liquid tanks [32]. A successful method to reduce the run-up as well as pressure peaks at the walls is the addition of additional structures inside the tank. This can be fixed inner structures [33–37] as well as potentially flexible elements [38]. Such a reduction of the impact of the sloshing is a next step planned for the floating cylinder.

This paper expands the previous experimental investigation and focuses on the mooring system to identify the potential influences on modelling assumptions. Therefore, different pre-tensions, wave tank deployment locations, as well as wave directions are investigated and discussed. An additional data set, which includes the wave direction of 180°, is provided via Edinburgh DataShare [39].

## 2. Methodology

### 2.1. Experimental Set-up

The experiment investigated a floating cylinder filled with water, and a similar solid ballast option, in the wave tank. It was identical to the system used for Gabl et al. [1,2,40] and extends the previously published results to provide a deeper understanding of the influences (e.g., mooring) on the interesting pitch/roll behaviour and other responses. The initial investigations compared the two ballast options with four different drafts under regular waves. The main goal was to identify the influence of the sloshing of the inner water body under large motions.

This experiment uses a single draft $d$ (the second lowest from the previous study) to further investigate the switch between the pitch and roll response, which only occurred for the water ballast configuration. A draft of $d$ equal 227.9 mm was chosen as it provides the clearest motions in the previous studies. The corresponding inner water level $h$ was 170.7 mm and the filled cylinder with an outer diameter $D$ of 500 mm had a total weight including ballast of 44.65 kg. The cylinder is open at the top but even the most extreme motion did not result in spilling or overtopping of the cylinder walls. Figure 1 presents an overview of the experimental set-up, which was investigated in the FloWave Ocean Energy Research Facility [41,42]. FloWave is a unique wave and current testing facility providing a full 360° capability for flow and waves. 168 wave makers are located in a circle (diameter of 25 m) and are used to generate as well as absorb waves. The upper volume has a constant water depth of 2 m and is separated by a 1 m thick floor (a centre section can be brought up to allow dry installation) with a similar lower volume. Both volumes are connected at the outside with 28 flow-drive units, which area also arranged in a circle. Currents of up to 1.6 m/s can be generated in any direction independently of the waves. Typical applications are the investigation of tidal turbines [43], novel velocity measurement approaches [44,45] as well as remotely operated vehicles [46]. For the presented investigation no current was used and the different wave directions were investigated without changing or rotating the experimental set-up in the tank.

Station keeping of the floating cylinder was achieved by four soft mooring lines. Two of them are connected to one point on the cylinder at the water line, so that the connection point is always at the height of the still water surface. This allows symmetry along the main axes in the tank but rotations around the $y$-axis were minimally influenced due to the low mooring stiffness in this degree of freedom (Figure 2). The previous experimental runs primarily investigated waves in the positive $x$-direction (90° in the tank definition; Figure 1) and this set-up influences the dominant pitch rotation to a lesser extent. The comparison of the solid and water ballast showed that the sloshing of the inner water led to a significant roll response of the water filled body. This paper aims to provide further insight into this effect and, when combined with the two available data sets [2,39] from the experiments in Gabl et al. [1], can be used as a validation experiments for numerical simulations focusing on Fluid-Structure-Fluid interaction with negligibly small deformations under big motions.

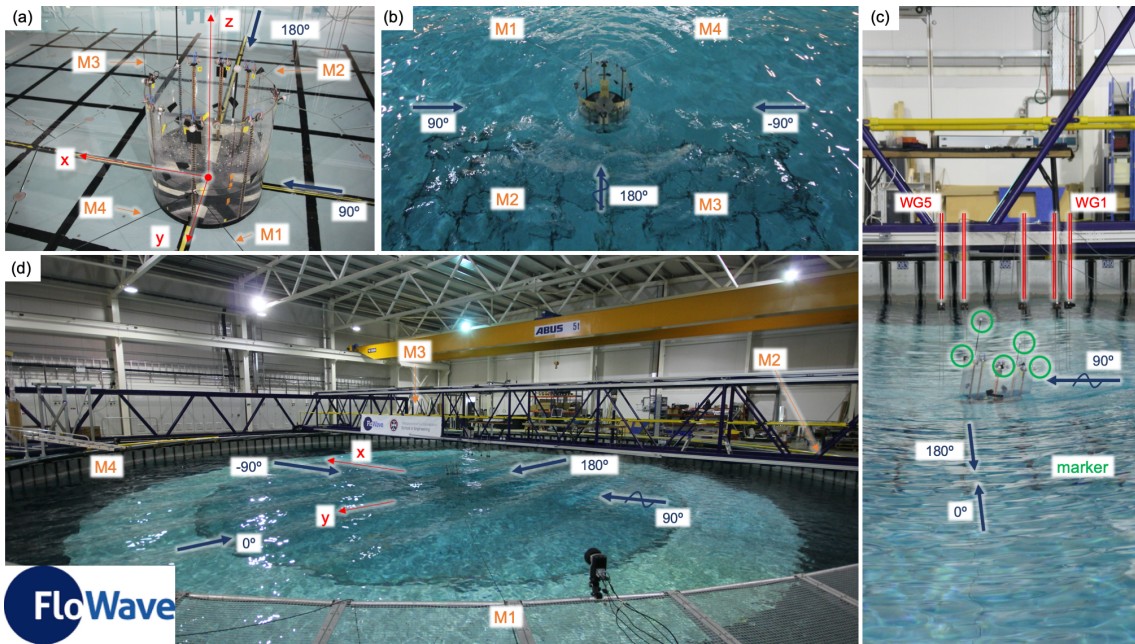

**Figure 1.** Experimental set-up in the tank on the raised tank floor (**a**) including the global coordinate system and the labelled mooring lines—(**b**) solid ballast case under waves coming from 180°—(**c**) side view of the experiments including the wave gauge (WG) array in the back—(**d**) overview of the full experimental setting in the tank.

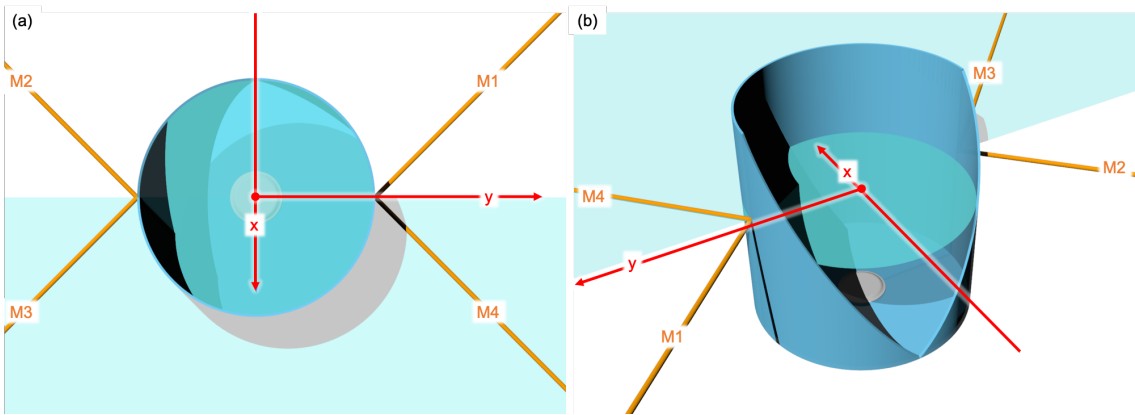

**Figure 2.** Rendered view of the cylinder from top (**a**) and from the side (**b**)—coordinate system moved due to better visibility.

## 2.2. Instrumentation

Two measurement instrumentation systems were used: (a) motion capturing system (MoCAP) and (b) wave gauges (WG). Synchronising these systems allowed the six Degree of Freedom (DoF) motion response of the floating cylinder as well as the free surface elevation to be documented. This system was similar to the previous experimental investigation [1,2].

The origin of the global coordinate system used by the MoCAP was located in the centre of the circular tank. A right-handed coordinate system was defined with a vertical $z$-axis. The $x$-axis was aligned with the main wave direction. As shown in Figure 1, this wave direction is defined as 90° in the tank definition. Therefore, the tank is split into two halfs with one side covering 0 to 180° and the other one with a negative sign. Both boundaries are identical. The eight Qualisys cameras (Göteborg, Sweden) of the upper water MoCAP were located at different heights on the tank side in an arc arrangement over approximately ±60°. Each camera captured the motion of the six markers mounted on the floating cylinder and the tracking is conducted with the software Qualisys Track

Manager (QTM, version 2019.3, Qualisys, Göteborg, Sweden). A rigid body was defined based on those markers and the 6-DoF motion calculated. The local body coordinate system used the same orientation as the global one and it was ensured that the origin is in the height of the still water surface of the tank. Regular refinement calibrations ensured a very high accuracy of the system, which was typical smaller 1 mm.

In the previous investigation, seven wave gauges (WG) were installed to cover a wide range in the *x*-direction of the tank. This provided an overview of the waves in front as well as behind the floating cylinder but no simultaneous values. For the current experiments, five WG were installed on the movable gantry in a reflection array with a constant offset in *y*-direction. The spacings between the WG were defined by a Golomb ruler with an order of 5 (marks (11, 9, 4, 1, 0); base length of 1 m; Table 1). Figure 1c shows the highlighted WG in the tank and the numbering in the main wave direction. The WG4 is located at $x = 0$ m, which is the initial position of the cylinder, unless otherwise stated. A regular 5 point calibration covering $\pm 100$ mm was conducted to ensure the high accuracy of this instrumentation, which is typically smaller than 1 mm [40,47].

**Table 1.** Location of the wave gauges (WG) in relation to the global coordinate system. The origin is in the centre of the tank and the spacing is defined by a Golomb ruler—outside diameter of the cylinder *D* equal 0.5 m.

| WGNr | WG1 | WG2 | WG3 | WG4 | WG5 |
|---|---|---|---|---|---|
| x (m) | −0.82 | −0.73 | −0.45 | 0.00 | 0.18 |
| y (m) | | | $-1.50 = 3 \cdot D$ | | |

The MoCAP used a sampling frequency of 128 Hz and the WG used 32 Hz. A digital pulse provided by the tank was used to synchronise both systems. In the majority of tests and if not otherwise stated, the capture time was 180 s and the periodic repeat time 128 s. The first 52 s covers the ramp-up phase, which was not included in the following presented results.

*2.3. Experimental Conditions*

All experiments were conducted with regular waves having a single wave direction. FloWave can provide a full 360° capacity for waves as well as for current but for this investigation, only wave directions between ±90 and 180° were used. Waves coming from the other half of the tank would have moved the floating cylinder in the direction of the WG, which could have led to a collision. It is assumed that the experimental setup is symmetric along the *x*-axis (Figure 2).

A constant requested wave amplitude $a_W$ (input value for the wave makers) of 50 mm was used (unless otherwise stated) and the requested wave frequency $f_W$ varied between 0.3 to 1.0 Hz. A Fast Fourier Transform (FFT) analysis was conducted for each WG and the maximum value of the amplitude spectrum was identified. The motion data were similarly analysed. Figure 3 presents a summary of the comparison of the requested and measured WG values for the investigated cases. The differences $\Delta a$ and $\Delta f$ were calculated by subtracting the requested value from the measured amplitude. Additional auxiliary lines are included to show representative percentages of the requested wave amplitude and frequency. Beside the five individual results for the WG, mean values over all WG are presented in Figure 3. This mean value is later used to calculate the RAO.

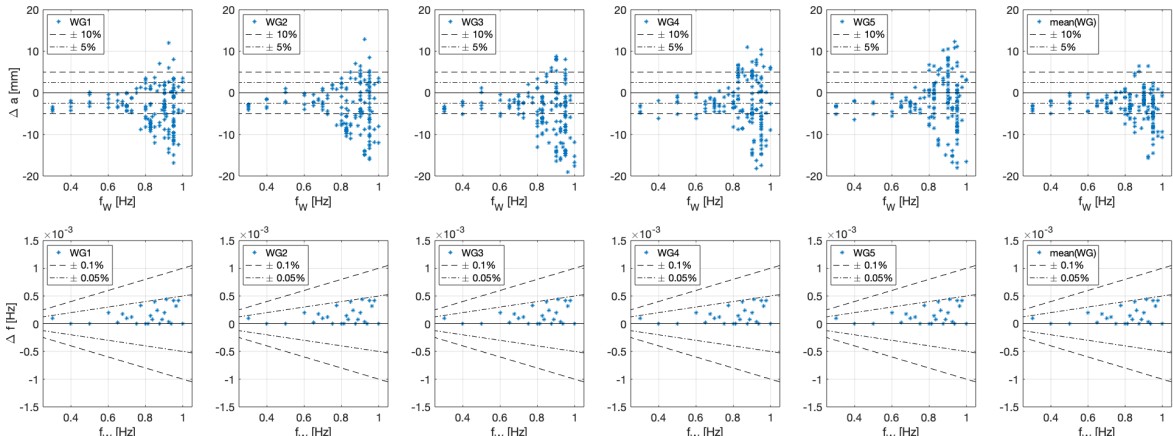

**Figure 3.** Analysis of the five individual wave gauge (WG) data, as well as the mean value conducted for the wave amplitude (**upper row**) and frequency (**lower row**)—difference $\Delta a = a_R - a_W$ (measured amplitude minus the requested amplitude); similar for the frequency.

The analysis of the wave amplitudes show that for the higher frequencies the spreading of the results increased. FloWave typically delivers reduced wave amplitudes [2,46], which are corrected as part of open tank calibration tests. The observed higher values are likely caused by reflections coming from the cylinder. This effect is smoothed by averaging over the five WG and these averaged values are used for the normalisation. The discrepancies in the wave frequency are very small ($<5 \times 10^{-4}$ Hz) and in the following sections the requested wave frequency $f_W$ is used.

## 3. Results

### 3.1. Overview

The presented research addresses four research questions (RQ): position of the cylinder in the tank (RQ1), length of the observation time (RQ2), sensitivity of the wave amplitude (RQ3) and wave direction (RQ4). In the first step, the sensitivity of the position of the floating object under wave conditions was investigated to ensure that the roll response was not caused by a very unlikely but potential local hotspot of reflection in the tank. This first RQ is investigated in Section 3.2, which also includes a change of the pre-tension of the mooring lines. Section 3.3 presents the results of the test investigating the influence of the length of acquisition time (RQ2). A sensitivity study was conducted for the wave amplitude and the results of RQ3 can be found in Section 3.4. The fourth RQ utilises the 360° wave capacity of FloWave and different wave directions are investigated (Section 3.5). The main focus was the 180° wave direction case, which was orthogonal to the previous experiments. For this research question, the solid ballast option was also investigated to provide a direct comparison between the two options for this additional wave directions.

All the presented investigations are limited to one draft of 227.9 mm (determined from the previous experimental programme [1,2]) with an inner water level $h$ of 170.7 mm and a total weight of 44.65 kg. Unless otherwise mentioned, the presented experiments were conducted with the water filled ballast option, a requested wave amplitude $a_W$ of 50 mm and under a wave direction of 90° based on the tank definition (Figure 1).

### 3.2. Position in the Tank

The first research questions focuses on the location of the floating cylinder in the wave tank, which has a diameter of 25 m. In the previous investigation the initial position in still water was always the tank centre and the regular waves pushed the cylinder in the positive $x$-direction. Large motions in surge were observed caused by the very soft mooring system. As a first step of the presented investigation the frequency sweep was repeated to test how repeatable the results are. The main focus

was on the occurrence of the roll motion. In the following figures the graphs labelled with *reference* uses the published data based on Gabl et al. [1,2] and *repetition* marks the similar experimental set-up conducted as part of this additional investigation.

The four individual mooring lines consisted of a hollow elastic of 3 m long (diameter 3 mm) with a very high stretch factor connected directly to the cylinder at two points, which could be adjusted so that it was always at the height of the still water surface of the tank. Each elastic was expanded with standard non-stretch rope to the tank side, which had a fixed length of 6.5 m. A small pre-tension was added to avoid the possibility of slack lines. For the *pre-tension* cases the rope was reduced by approximately 3 m for all mooring lines. Hence, the initial position stayed the same but the pre-tension in the mooring lines was increased. By releasing two mooring lines, namely M3 and M4 (Figures 1 and 2), the initial position of the cylinder was moved into the negative $x$-direction. Similarly, the mooring lines M1 and M2 was brought back to the original condition to investigate the initial motion in the positive $x$-direction. For these tests the wave direction was changed to $-90°$, which brought the cylinder approximately to the same relative location under wave loads. The definition of the DoF are not changed for those specific cases with a different wave direction.

Figure 4 presents the mean values of the repeat time for the previously described cases in relation to the wave frequency $f_W$. The difference of the modified set-ps can be seen in the surge analysis. The *reference* and *repetition* data as well as the increased *pre-tension* show similar response starting from 0 for the lower frequencies and up to around 1.5 m around 0.8 Hz. A nearly constant offset can be observed for this analysis due to the change of the initial position the floating cylinder in the tank. For frequencies around 0.8 Hz the floating cylinder reaches the centre of the tank and the WG array covers this range (Table 1, Figure 1). The limited cases conducted with an initial motion in positive $x$-direction show a symmetrical behaviour around $x = 0$ m. The differences for all other DoF are relatively small. Nevertheless, the mean value of pitch reaches approximately up to 5°, which is not detected by the corresponding FFT when examining the maximum value of the amplitude spectrum. The *moved* $x^+$ cases show a negative mean pitch value indicating that the wave direction for those cases is in the opposing direction and the definition of the DoF was not changed.

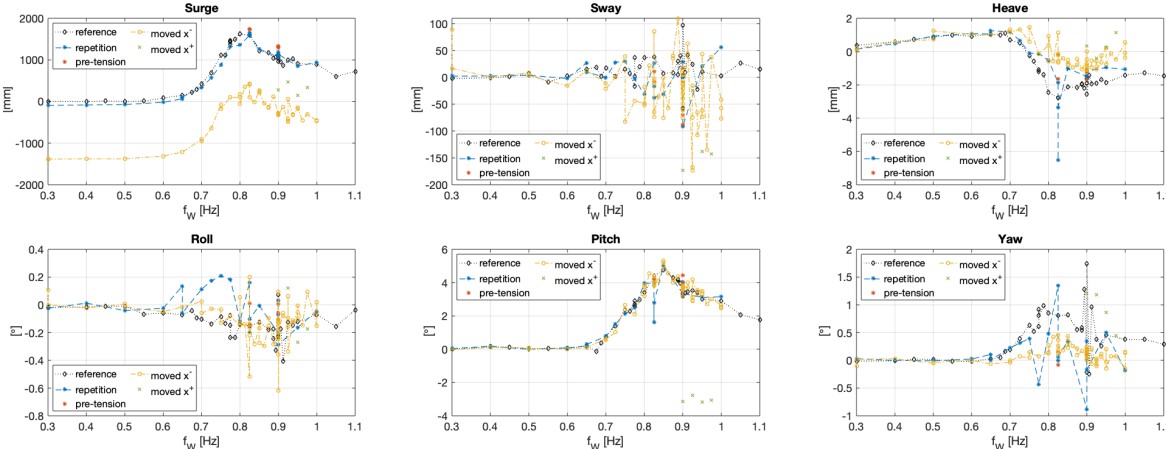

**Figure 4.** Average value of the repeat time for all DoF—the reference measurement uses data from Gabl et al. [1] and is compared to the repetition of those experiments as well as changes of the pretension and different initial position in the tank.

The maximum amplitude response based on the FFT analysis is presented in Figure 5 and the corresponding response frequencies in Figure 6. Both analyses are presented in relation to the wave frequency $f_W$. The surge and sway responses are larger for the *moved* $x^-$ but the response frequencies for this frequency band are small. It is highly likely that this was caused by the asymmetric pre-tension in the mooring lines (M1 and M2 was larger than M3 and M4, Figure 1). The other DoF do not show significant differences considering the unequal distribution of the pre-tension in all four mooring

lines. The heave motion shows nearly no difference between the investigated cases. Considering the rotations, the *repetition* generally resulted in a slightly smaller response in comparison to the previous experimental result.

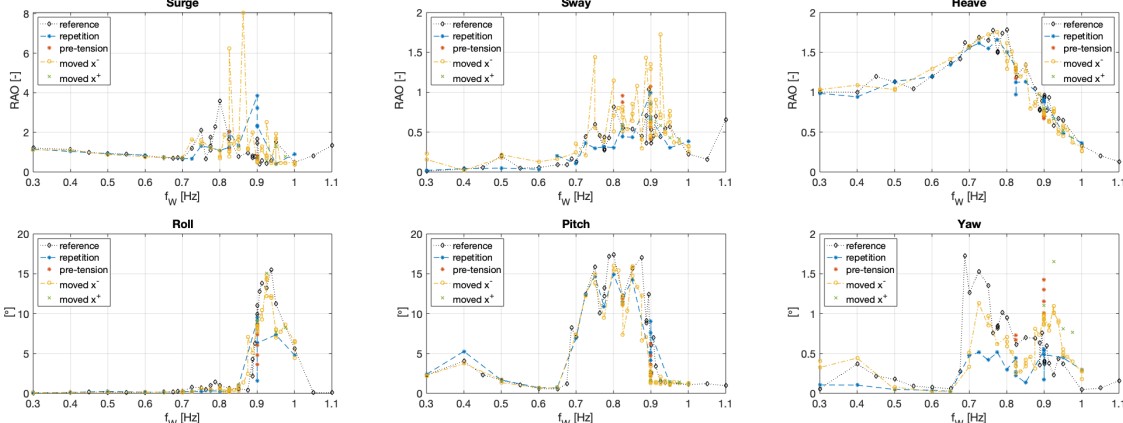

**Figure 5.** Amplitude response in the six DoF in relation to the requested wave frequency $f_W$.

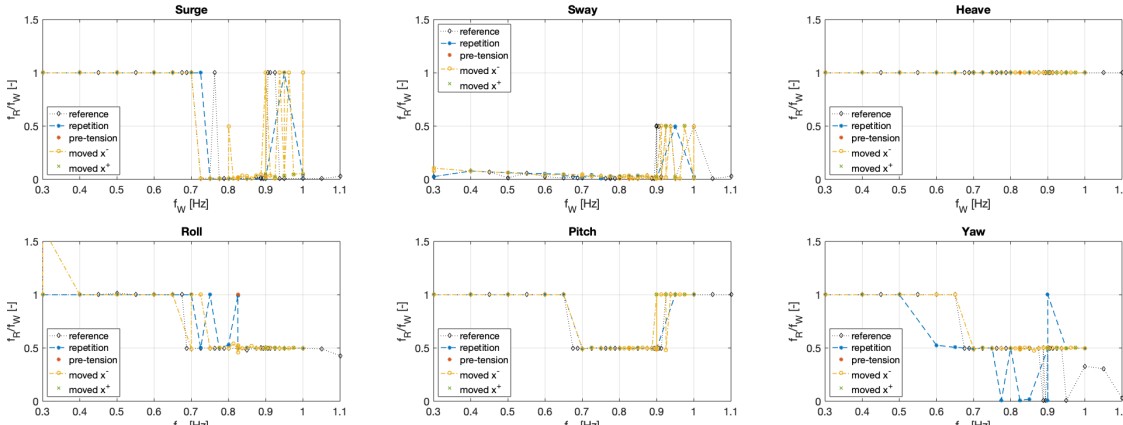

**Figure 6.** Frequency response $f_R$ normalised by and presented in relation to the requested wave frequency $f_W$ for the amplitude results in Figure 5.

In addition to the overview in Figure 5, the responses in roll and pitch are analysed in Figure 7 for the individual cases with adapted scaling of the *y*-axis (wave frequency $f_W$). The first set of graphs show the comparison between the results of the *reference* and *repetition* measurements. The switch between roll and pitch happens at a frequency of approximately 0.9 Hz and therefore eight repeat tests were conducted for this case. In general a relatively good agreement with the previous investigation could be found, which is in the range of the differences between different repetitions (back to back testing). The largest differences were observed for the wave frequency band with a significant response in pitch and roll, which is discussed in Section 3.3. The higher frequency band could be further investigated to provide more points in this comparison. Nevertheless, the very similar pitch response shows a good agreement and this frequency band was further investigated as part of the *moved $x^-$* cases. This comparison is shown on the right side of the Figure 7. The changed *pre-tension* as well as the *moved $x^+$* cases result are very similar to the other results.

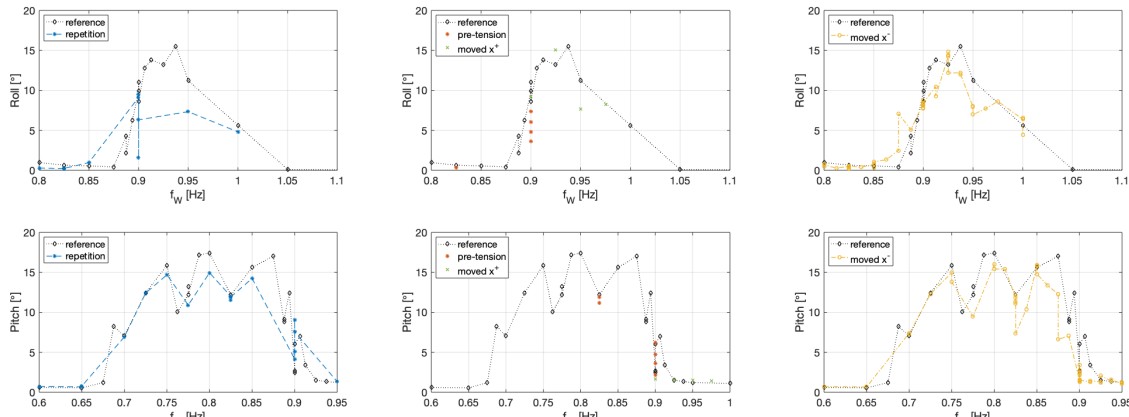

**Figure 7.** Detailed view of roll (**upper row**) and pitch (**lower row**) presented in Figure 5—the used boundaries for wave frequency $f_W$ are limited to focus on a specific frequency band for each DoF.

The analyses presented in Figures 5–7 investigate the pre-tension and location of the floating cylinder in the tank. All cases showed a repeatable occurrence of the roll motion response and consequently the RQ1 can be answered in that the roll response is not depending on the specific tank position. Furthermore, it can be summarised that the results for the majority of wave frequencies show no significant influence of these variables. For the transient frequency band the results can vary significantly between similar test and this instability is further investigated in Section 3.3. The largest differences are caused by the different pre-tension in the mooring lines or occur with at a small response frequency.

### 3.3. Length of the Test

As mentioned in Section 3.2, the largest differences between the *reference* data and the *repetition* could be found in the wave frequency $f_W$ band around 0.9 Hz. The switch between roll and pitch is observed around this frequency for the wave direction of 90° in the tank definition. As shown in Section 3.5, this value is significantly reduced by changing the wave direction to 180°.

The second research question focuses on the length of the capture time. A typical total length of 180 s was used. To ensure that no effects are missed, extended experimental runs were conducted. Therefore, the length is tripled from 180 s (3 min) to 540 s (9 min). Considering a similar ramp-up time of 52 s, the extended observation time contained 488 s. Figure 8 presents the time series of roll and pitch for six examples. The ramp-up time is marked with a red vertical line and the horizontal lines provide the results of the FFT analysis as a response amplitude around the mean value of the repeat time.

The first three time series show a wave frequency $f_W$ of 0.9 Hz with one normal and two repetitions of the extended run. Comparing the chosen ramp-up time for these cases show that the floating cylinder first responds with a pitch rotation. Roll starts later and when it occurs pitch is reduced. The short run shows that an extended ramp-up time could potentially improve the analysis. But the comparison at the same wave frequency with an extended observation time (second and third set in Figure 8) show that such a switch between roll and pitch can also occur later in the experiment. The repetition show that only the basic behaviour can be reproduced. A splitting of the complete time series in sub-section could improve the analysis but the main finding is that the results in this transition zone show a large variance of responses. Consequently, this frequency band should be used primarily for a qualitative and not quantitative comparison of numerical results. A similar comparison for the $f_W$ equal 0.925 Hz is also presented in Figure 8. This frequency results in a clear roll response starting with a very small initial pitch response. An extension of the observation time proves that the stable behaviour is not changing. Similar results are shown with the further increased wave frequency of 0.9375 Hz (last set in Figure 8).

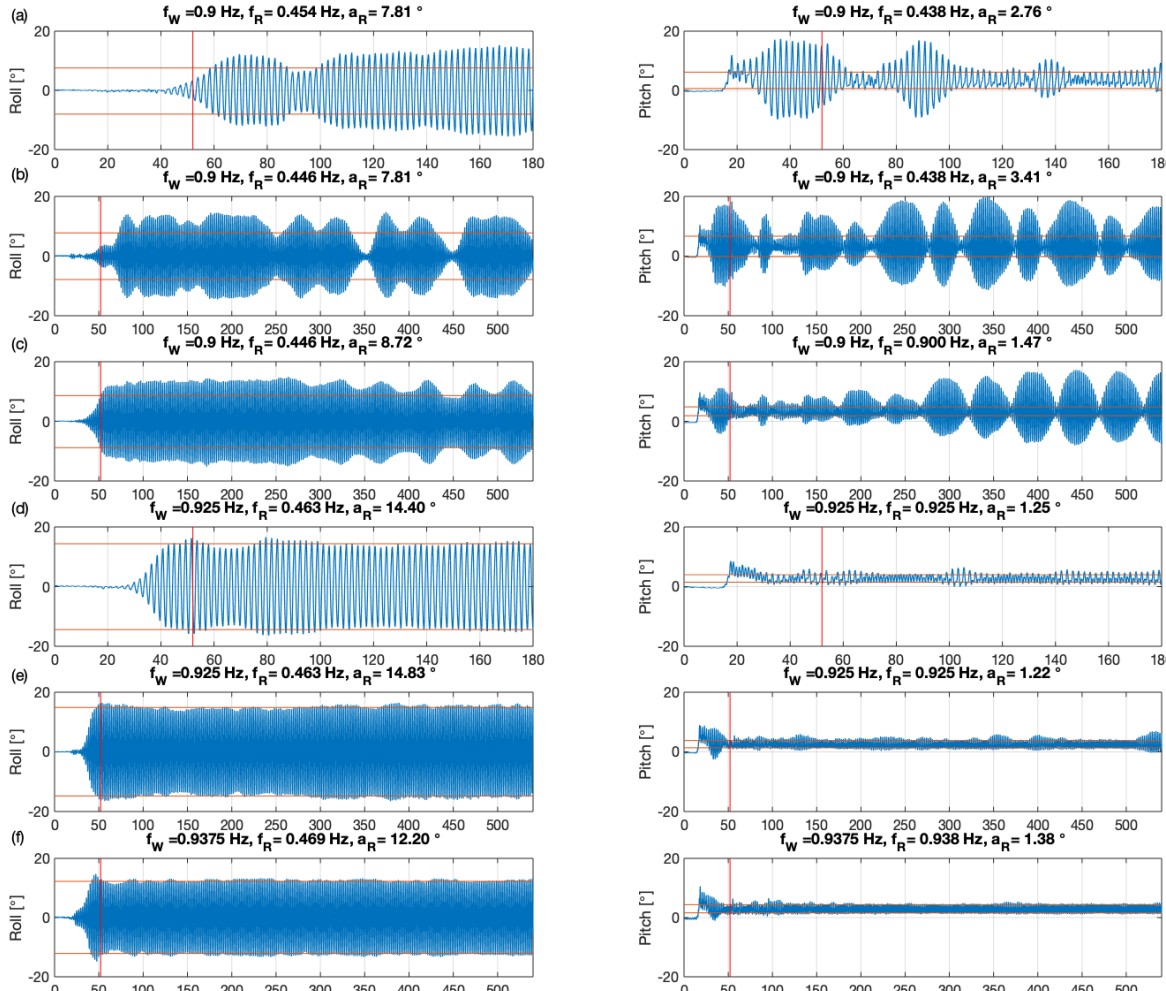

**Figure 8.** Sample time series of roll and pitch for six test runs with different requested wave frequency $f_W$ and expanded capture time—(**a**) $f_W$ = 0.90 Hz, 180 s, (**b**,**c**) 0.90 Hz, 512 s, (**d**) 0.925 Hz, 180 s, (**e**) 0.925 Hz, 512 s, (**f**) 0.9375 Hz, 512 s—vertical red line indicates the 52 s rump-up time. The horizontal red lines mark the response amplitude $\pm a_R$ and the connected response frequency $f_R$ is provided in the title of the individual graph.

In addition to this individual time series, Figure 9 presents additional analysis for those two rotation responses, while the first 52 s are coloured in red. Therefore, the pitch response is plotted against the measured roll values in the right column of Figure 9. The oscillation shows a positive average pitch value, which can also be seen in Figure 4. The results for a wave frequency of 0.9 Hz show a butterfly like response covering both rotation directions. Outside of this transient zone a clear response can be observed, which for the additional three wave frequencies takes the form of a significant roll response. In the other two columns of Figure 9 the specific angle is compared to the corresponding velocity. A simple harmonic oscillator would show a circular behaviour with the maximum speed at an angle equal to zero. Only for one repetition of the wave frequency 0.9 Hz could such a comparable circular behaviour be found and the response of both DoF are in the same magnitude. A clear pattern of circular behaviour can be found for cases with a higher frequency and a dominant roll response. The response in pitch is similar and it has to be considered that this would be obvious response direction based on the incident waves.

Based on the conducted comparisons and the RQ2, it can be stated that the 180 s capture time is sufficient. In the transient zone a non-stable switch between pitch and roll could be observed, which complicates the quantification of the separate rotations and reduces the RAO results to a very

rough assumption. Consequently, it is advisable to use a frequency band with a clear response for future validation of numerical simulations.

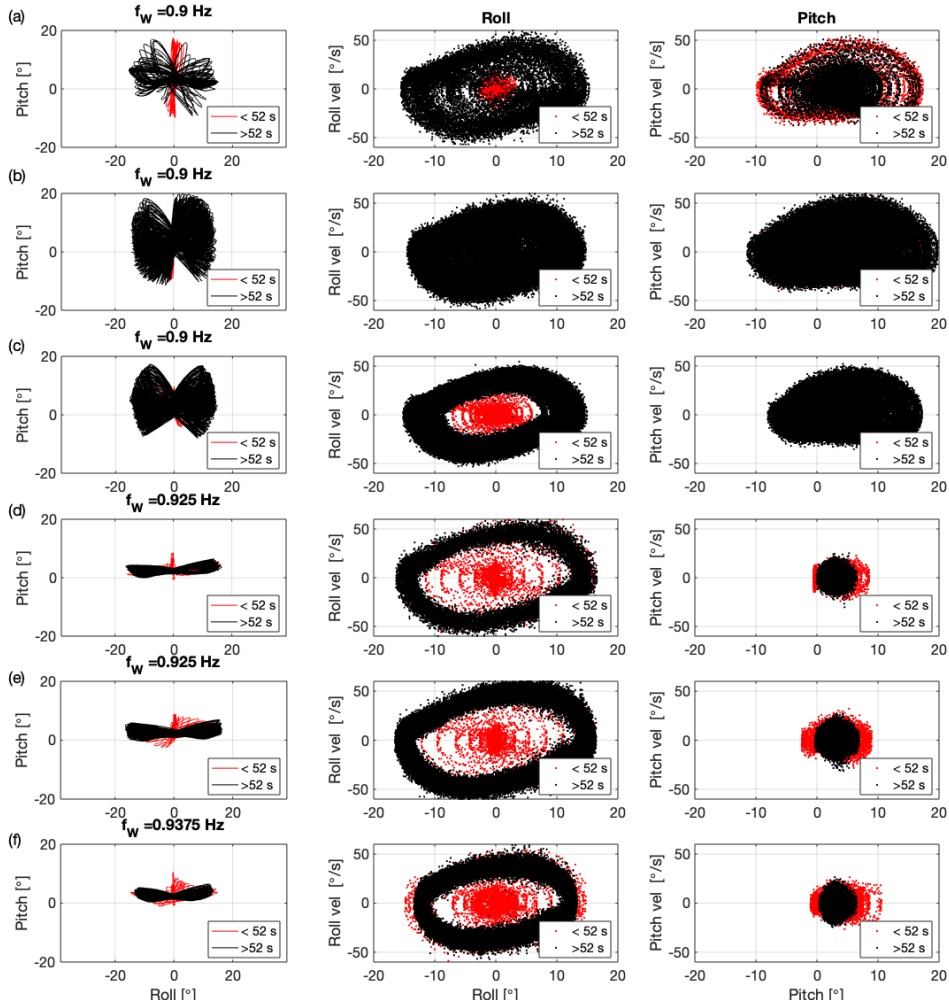

**Figure 9.** Direct comparison of pitch and roll as well as the comparison of pitch and roll angle with the corresponding velocity for the time series presented in Figure 8—(**a**) $f_W$ = 0.90 Hz, 180 s, (**b**,**c**) 0.90 Hz, 512 s, (**d**) 0.925 Hz, 180 s, (**e**) 0.925 Hz, 512 s, (**f**) 0.9375 Hz, 512 s—the first 52 s are presented in red.

### 3.4. Wave Amplitudes

Early in the research project and based on unpublished preliminary experiments the decision was taken to run the experiments with a constant requested wave height of 50 mm. The intention is that the amplitudes should be large enough to produce a measurable response without being so large as to negate any assumptions of linear behaviour required for numerical modelling validation. The main comparison of water and solid ballast options is based on this assumption [1,2]. As part of this additional experiments, this decision was again tested. It has to be mentioned that this was not a comprehensive investigation and the main aim is to understand how far the results are independent of the wave amplitude. This is important if numerical codes simulating in the frequency domain are be used to reproduce the experiments.

Figure 10 presents the response amplitudes for all six DoF for the conducted experiments with different requested wave amplitudes and Figure 11 the corresponding response frequencies $f_R$ normalised by the wave frequency $f_W$. Both figures use the measured wave amplitude $a_R$ for the $x$-axis of the graph. The results are presented as Response Amplitude Operators (RAO) and grouped by $f_W$ and wave direction.

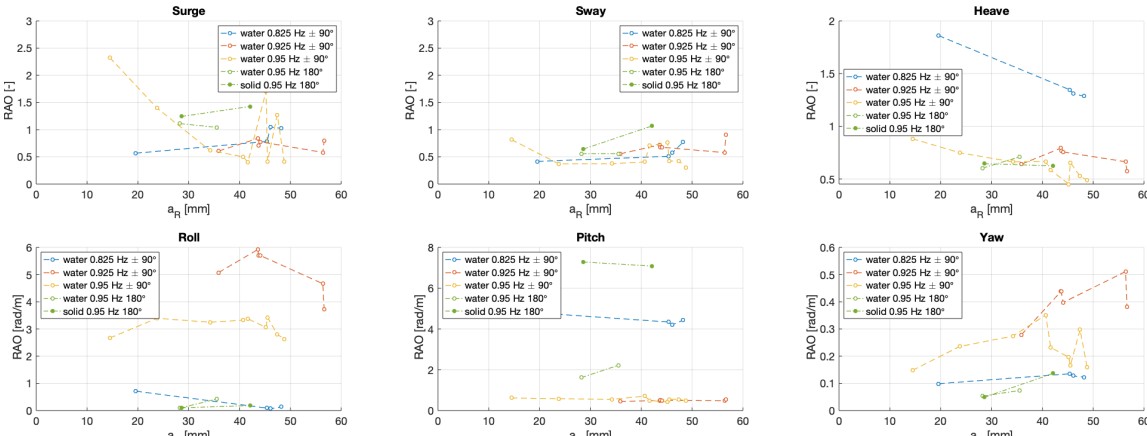

**Figure 10.** Amplitude response in the six DoF in relation to the measured wave amplitude $a_R$—individual graphs with a fixed requested wave frequency $f_W$ and wave direction

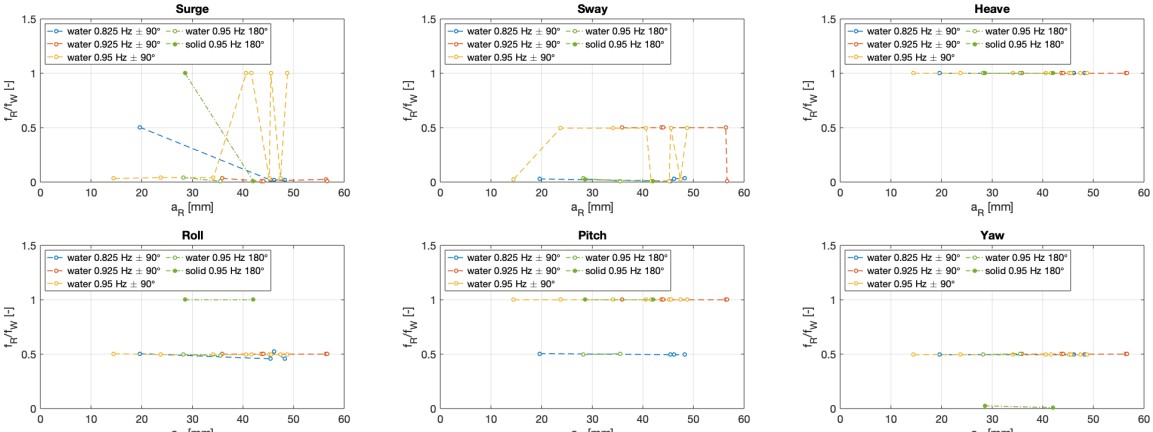

**Figure 11.** Frequency response $f_R$ normalised by the requested wave frequency $f_W$ and presented in relation to the measured wave amplitude $a_R$ in addition to Figure 10.

The motion responses show a relatively constant response. The bigger jumps in surge are connected with changes in the response frequency. Lower wave amplitudes show larger differences to the other investigated amplitudes. This is especially noticeable for the 0.825 Hz wave case, which is very close to the peak response in heave. Further experimental runs are needed to fully investigate this.

The research question RQ3 focuses on the sensitivity of the wave amplitude. The limited cases show that the differences for the motion RAO are mainly in the range of 0.5. This range expands to typically 1 rad/m for the rotations. A relatively constant response for roll and pitch can be observed and the yaw components are relatively small. Further experimental investigations including a wide range of wave amplitudes are needed to fully assess this.

### 3.5. Direction Including Comparison with the Solid Case

The main focus of this investigation lies in the change of wave direction. Hence the mooring system is symmetrical around the $x$ and $y$-axis, waves coming from the orthogonal direction are particularly interesting. Previously, the main wave direction was 90° in the tank direction and along the positive global $x$-axis. As shown in Figure 1, two mooring lines were connected in one point, which allowed a smaller influence for the rotation around the $y$-axis (pitch for 90°). Rotations around the $x$-axis (roll for 90°) were more influenced by the mooring system. Consequently, it was assumed that the observed switch from pitch to roll for the water ballast case was suppressed rather than caused by the mooring. This was tested as part of the current experimental investigation by rotating the wave direction to 180° (Figure 1). If not otherwise stated, the local body coordinate system was aligned

with the wave direction. Hence, roll is defined as rotation around an axis in the direction of the waves and pitch orthogonal to it. The following presented investigations were also conducted with the solid ballast option to provide the direct comparison for the additional wave direction. The plotted cases with water ballast are indicated with an empty marker and a solid one for the experimental runs with a solid ballast.

Figure 12 shows the summary of the additional comparison and focuses on the key three DoFs, namely roll, pitch and heave. The corresponding frequency response can be found in Figure 13. Both figures also include the reference values for the solid ballast option. Those experimental results were measured under a wave direction of 90° and good reproducibility of the occurrence of the roll response is shown in Section 3.2. A full frequency sweep was conducted with the 180° wave direction as part of the additional experimental investigation as well as additional wave runs in ±135°. For the latter, it has to be highlighted that the investigated mooring system was symmetrical along both main axes, but not identical—two mooring lines are connected in one point on both sides of the cylinder. For waves along one mooring line the system is not balanced and the mooring system influences the motion response significantly. Those results were not ideal, nevertheless they are included in the paper to show why only the main directions are considered in this study for this mooring configuration.

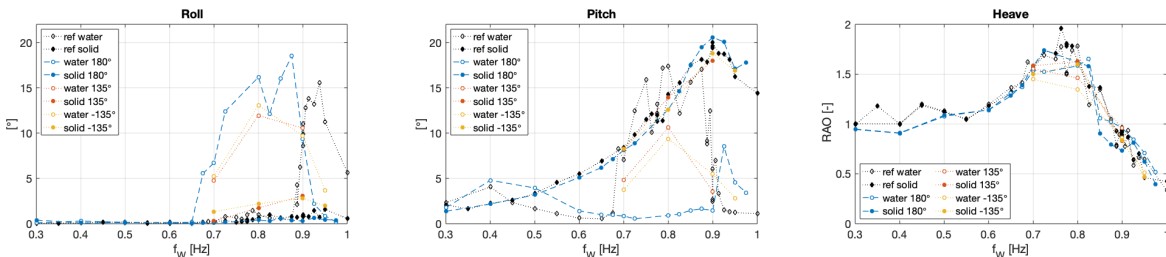

**Figure 12.** Roll, pitch and heave response for the wave directions 180° and ±135° compared for the references data (90°; [1])—water ○ and solid ● ballast option.

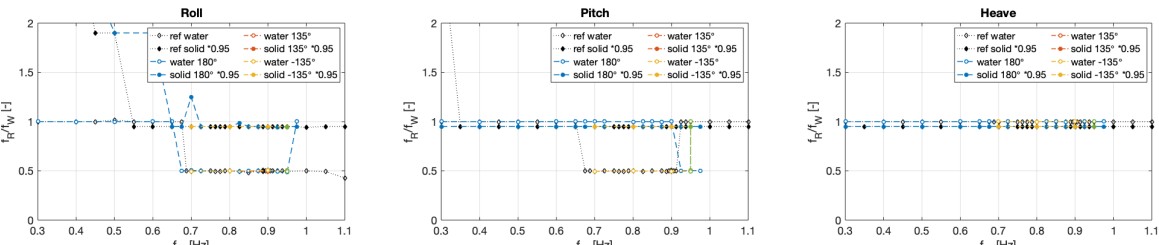

**Figure 13.** Frequency response $f_R$ normalised by the wave frequency $f_W$ in addition to Figure 12—water ○ and solid ● ballast option, which is multiplied by 0.95 to allow a better visibility of the results.

The response in the vertical direction (heave, Figure 12) is very similar for all investigated cases, which was already shown in Gabl et al. [1]. As presented in Figure 13, the response frequency for heave is identical for all wave frequency $f_W$ (the values of the solid cases are multiplied by a factor of 0.95 so that the overlay of the curves is reduced). Changing the wave direction consequently has no influence on the heave motion.

The obvious difference in the motion response between the two investigated wave directions, namely 90 and 180°, can be found in the roll response (rotation around the axis in wave direction) and for the water filled case. In the *reference* investigations, the floating cylinder responded in the lower frequencies with a pitch response. Crossing a transition frequency band (Figure 8), a further increase of the frequency results in a roll oscillation. This can only be observed for the water filled cylinder and consequently is caused by the sloshing of the inner water body. By changing the wave direction to 180°, the principal main rotation stays very similar. Hence, for this wave direction the cylinder responds for a wide frequency band with a roll motion instead of pitch. In the higher frequency, where roll previously occurred, rotations are reduced and a small response in pitch can be observed. The response

frequencies in roll (Figure 13, left graph) are very similar for both wave directions. A reduction to 0.5 for $f_R/f_W$ can be seen over a $f_W$ of 0.65 Hz. This indicates that the potential for the roll response is present in both wave direction but the weaker response of the mooring system is around the global $y$-axis, which leads to a pitch motion in the 90° case. The pitch response of the 180° remains relatively small in the lower frequency band and is mainly in the same frequency as $f_W$. This is in contrast to the 90° case, which response at half $f_W$ with relatively large pitch values.

Figure 14 presents three pair of time series for waves with the similar wave frequency $f_W$ of 0.85, 0.9 and 0.95 Hz for both wave directions (90° and 180°), and Figure 15 shows the detailed analysis of the two rotation around the $x$ and $y$-axis, which is comparable to Figure 9. In this particular case the measured rotations around the global axis are presented. Hence, the left column in Figure 14 includes the results for the rotation around the $x$-axis and the right column the one around $y$-axis (definition presented in Figure 1). Based on the actual wave direction, roll and pitch are mentioned in the label of the axis. The first row shows the results of the 0.85 Hz wave along the global $x$-direction (90°). After the waves reach the floating cylinder an initial increase in pitch is observed and a stable rotation around $y$ builds up. Only a very small roll motion is visible. The same wave train is used for the 180° direction presented in the second row of Figure 14. An identical initial motion is observed for the rotation around the $x$-axis (the different signs are caused by the definition of the positive direction; in both cases the top of the cylinder moved first in the wave direction). This initialised oscillation in the wave direction (around the $x$-axis) switches to a response around $y$.

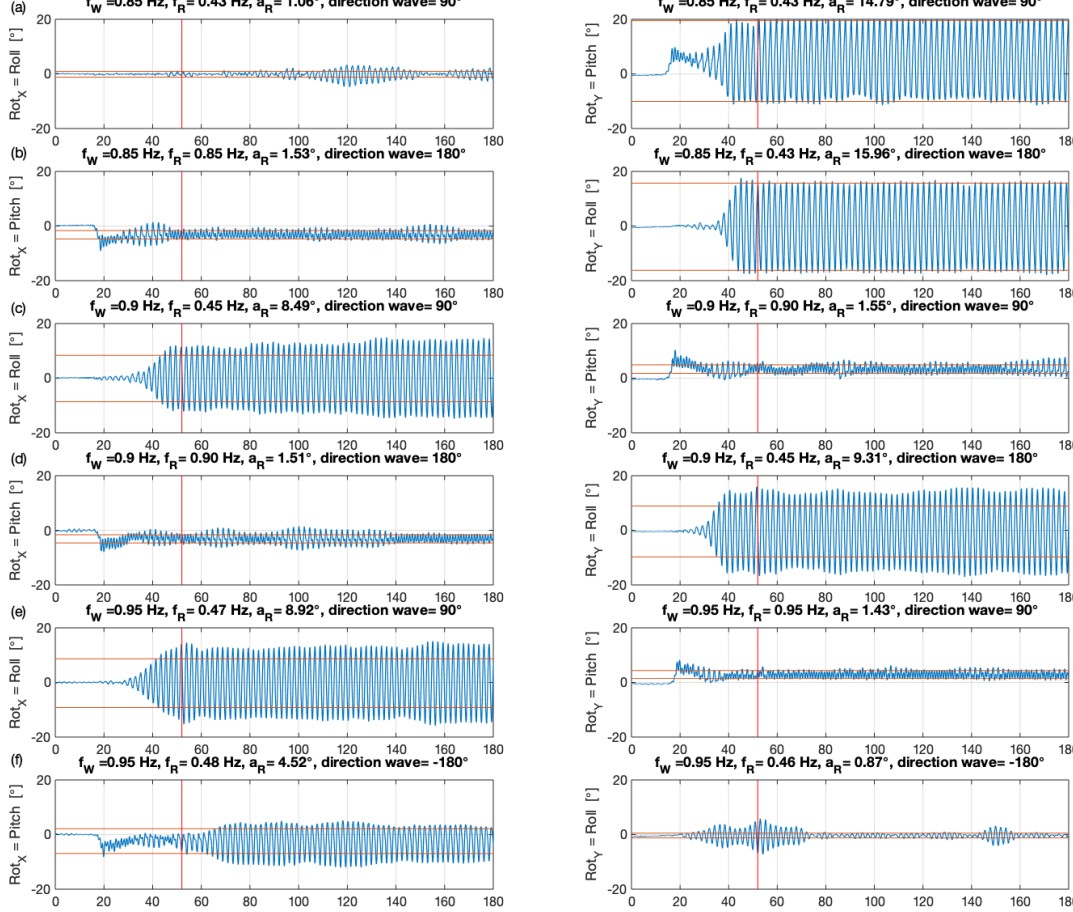

**Figure 14.** Sample time series of roll and pitch for six test runs with different requested wave frequency $f_W$ and comparison of the wave direction of 90° and 180°— (**a**) $f_W$ = 0.85 Hz, 90°, (**b**) 0.85 Hz, 180°, (**c**) 0.90 Hz, 90°, (**d**) 0.90 Hz, 180°, (**e**) 0.95 Hz, 90°, (**f**) 0.95 Hz, −180°—vertical red line indicates the 52 s ramp-up time. The horizontal red lines mark the response amplitude $\pm a_R$ and the connected response frequency $f_R$ is provided in the title of the individual graph.

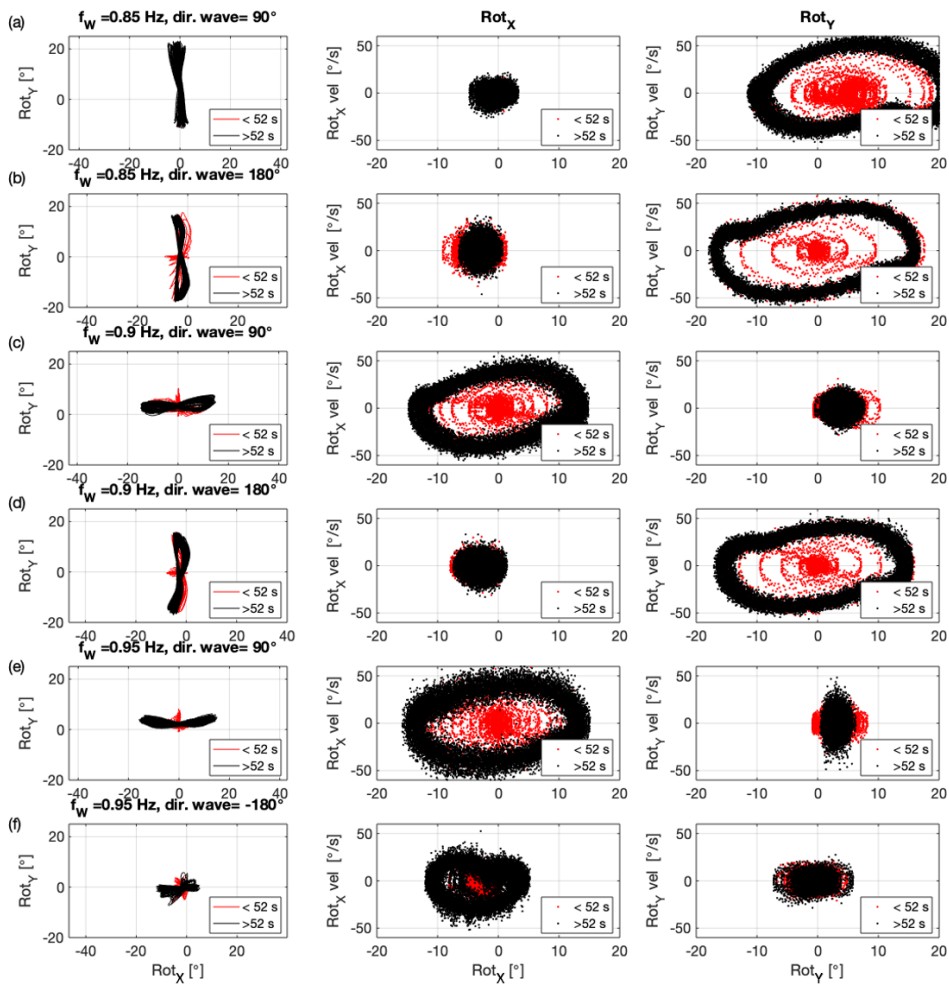

**Figure 15.** Direct comparison of pitch and roll for the time series presented in Figure 14—(**a**) $f_W$= 0.85 Hz, 90°, (**b**) 0.85 Hz, 180°, (**c**) 0.90 Hz, 90°, (**d**) 0.90 Hz, 180°, (**e**) 0.95 Hz, 90°, (**f**) 0.95 Hz, −180°—the first 52 s are presented in red.

A full change of the dominant rotation with the wave direction can be found for the wave frequency of 0.9 Hz. By further increasing the frequency to 0.95 Hz, a comparable behaviour to 0.85 Hz could be observed. The roll response for the 90° cases remains in the same direction for the 180°, which is now pitch.

As mentioned previously, only the main direction along the global horizontal axis provides an unbiased response of the floating cylinder, hence the mooring system does not possess a rotational symmetry (Figure 1). Figures 12 and 13 include results for some sample frequencies with wave directions of ±135°. The solid ballast option response is very close to the other wave directions but also a small additional rotation in roll can be observed. This influence of the mooring lines are larger for the water filled cylinder and makes it hard to extract an exact response. Nevertheless, a further variation of the wave direction was conducted for a constant wave frequency of 0.95 Hz for both ballast options. Figure 16 summarises the amplitude responses in relation to the wave direction and the associated response frequency $f_R$ is provided in Figure 17. The influence of the mooring system can be seen in the small increase of the roll response for the solid version between −180° (equal to +180°) and −90°. Overall similar range, the response amplitude for roll decreases approaching −180°, which was also observed for the last pair of time series with a wave frequency of 0.95 Hz presented in Figure 14. The pitch response is relatively large in relation to the other wave directions. This can also be clearly observed in the response frequency $f_R$ in pitch at a ratio of 0.5 in relation to the wave frequency $f_W$ (Figure 17). The change in heave is small.

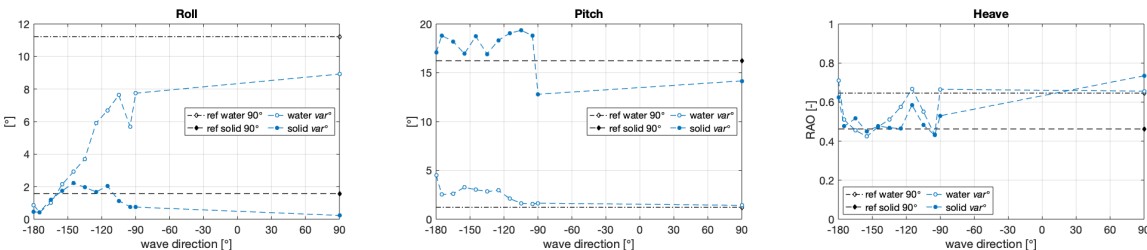

**Figure 16.** Roll, pitch and heave response for different wave directions with a fixed wave frequency of 0.95 Hz—water ○ and solid ● ballast option.

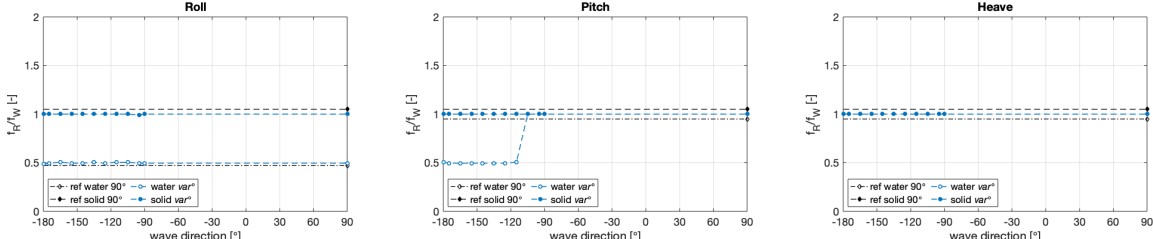

**Figure 17.** Frequency response $f_R$ normalised by the wave frequency $f_W$ in addition to Figure 16—the reference values (either 0.5 or 1[-]) are multiplied by a factor to make them better visible—water ○ and solid ● ballast option.

The key finding of the variation of the wave direction and the research question RQ4 is that the chosen mooring system is not the cause of the roll response and more likely prevented it. A change of the wave direction to 180° results in a significant increase of the frequency band associated with a roll response, while the solid ballast option showed almost no dependence on the changed wave direction. Consequently, those results are a vital expansion of the previously published data set and is available as an addition via Edinburgh DataShare [39].

## 4. Discussion

The chosen mooring system and potential alternatives are discussed below. Furthermore, the changes in the position of the wave gauges (WG) from the initial layout, presented in Gabl et al. [1,2], to the current one are highlighted and a brief overview of future work provided.

The mooring system was introduced as part of the previous experimental investigation exploring four different drafts of the floating cylinder with water and solid ballast. The station keeping mooring configuration deliberately provided a smaller resistance for the rotations around the $y$-axis to ensure a minimum of influence for this degree of freedom (Figure 2). This asymmetry was intended to provide a maximum freedom for rotations in the wave direction and a slightly higher resistance in the orthogonal one. This lower resistance occurred for the roll response by turning the wave direction by 90°, which lead to a significant increase of the frequency band with a roll response. Consequently, the rolling of the floating cylinder is more likely to be restricted by the mooring configuration, rather than caused by it. A higher degree of symmetry could be achieved by connecting each mooring line to one point on the cylinder. Such a system would be symmetrical for all four quadrants but also lead to two different main directions, namely along a mooring line as well as between two mooring lines. Ideally a fully rotationally symmetric mooring system would be desirable. One option to achieve this could be a single mooring line connection at the bottom of the cylinder. A disadvantage of this approach is, that the introduced mooring force might have an increased influence on the motion response of the cylinder. It could also introduce a greater self-aligning torque and prohibit the oscillation. The chosen system was a good compromise between station keeping and allowing the motion response to develop freely. Nevertheless, a complete change of the mooring system could be a further step forward including a variation of the mooring line stiffness. Alternative approaches for mooring systems, which might restrict different DoF will also be considered.

For the previous experimental runs the wave gauges (WG) were distributed over a wide range in front as well as behind the floating cylinder. This allowed the provision of a very good measurement of the incoming waves and potentially changes caused by the cylinder. A direct comparison of the motion response with the WG was not possible. For the new experiments, the WG were arranged with a *y*-offset to the floating cylinder (Figure 1). Especially in cases with a moved initial position (Section 3.2), a very good side by side alignment could be reached. The distance between the floating cylinder and the WG array was chosen to be 1.5 m, which is 3 times the diameter of the cylinder *D*. The results of the measured free surface elevation show bigger influences of reflections caused by the floating body, which has to be considered in future usage of the data as a validation experiment. A further translation of the WG in the *y*-direction would have reduced this influence. This would also allow the use of the full 360° capacity of FloWave without risking a collision of the cylinder with the WG. Furthermore, the WG remained at the same position for the different wave directions. Future experiments will include a adjustable WG array mounted on a rotatable arm. The alignment can be guaranteed with the MoCAP.

As previously mentioned in Section 3 an expansion of the investigated wave amplitudes as well as a further refinement of the wave frequencies is desirable. A further aim of the research project is to measure the changes of the free surface of the inner water body [40] as well as the velocities of the sloshing water body inside of the cylinder.

## 5. Conclusions

This work extended the work of Gabl et al. [1] in exploring the influence of the water ballast in a floating cylinder. In those original tests a pronounced roll response was observed towards the higher end of explored wave frequency range. This result was reproduced in these tests and is confirmed as a real effect.

It could be shown that the tests are reproducible and that the pre-tension of the soft mooring lines as well as the position of the floating cylinder in the tank have only a negligible influence on the response of the floating cylinder. An extension of the capture time proves that the 180 s minus the 52 s ramp-up time are sufficient to ensure that the motion response is fully developed and stable. In a transition frequency band the response between pitch and roll switches constantly and similar wave conditions can lead to varying results. Consequently, those frequencies should be avoided for validations experiments which target the investigation of the roll motion of the water filled structure, or at least this instability should be kept in mind. The variation of the requested wave amplitude from the standard value of 50 mm were limited to some exploratory cases but resulted in relatively stable responses. Nevertheless, for a direct comparison in the time domain the actual measured wave amplitude should be used.

A significant change could be observed for the changed wave direction. In contrast to the solid ballast option, the roll motion of the water filled cylinder started with a far smaller wave frequency and was the dominant rotation response for this case. This is caused by the mooring design, which allowed a smaller resistance for the roll rotation in case of the wave direction in 180°. Based on this result, it can be stated that the initial roll response of the water filled cylinder was not caused by the mooring system. It quite contrary prohibited it for the 90° direction. Consequently, the 180° wave direction is especially interesting for validation experiments aiming to reproduce the roll response of the water filled structure.

**Author Contributions:** R.G., T.D. and D.M.I. are responsible for the conceptualisation of the experimental investigation. R.G. measured the data and analysed the data. R.G. and T.D. wrote the initial draft and D.M.I. reviewed and edited the paper. All authors have read and agreed to the published version of the manuscript.

**Funding:** This work was supported by the Austrian Science Fund (FWF) under Grant J3918.

**Conflicts of Interest:** The authors declare no conflict of interest.

## Notation

| | |
|---|---|
| $a_R$ | amplitude waves (mm) measured |
| $a_W$ | amplitude waves (mm) requested from the wave makers |
| $d$ | cylinder draft (m) |
| $D$ | cylinder diameter (m) |
| $f_R$ | response frequency (Hz) measured |
| $f_W$ | frequency wave (Hz) requested from the wave makers |
| $h$ | water depth inside the cylinder (m) |
| $H$ | height of the cylinder (m) |
| $x$ | distance (m) in the main wave direction defined as 90° |
| $X$ | motion in $x$-direction (mm) |
| $y$ | distance orthogonal to the main wave direction (m) |
| $Y$ | motion in $y$-direction (mm) |
| $z$ | distance vertical direction (m) |
| $Z$ | motion in $z$-direction (mm) |
| DoF | degree of freedom |
| MoCAP | motion capturing system |
| RAO | response amplitude operator |
| RQ | research question |
| WG | wave gauge |

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
