# Peer review of "Roll Motion of a Water Filled Floating Cylinder—Additional Experimental Verification"

_water, doi:10.3390/w12082219_

Round 1
Reviewer 1 Report
Please see attached document for comments.

Author Response
We thank the reviewer for his/her comments and suggestions. A separate document with the response to all three reviewers is provided. Thank you.

Reviewer 2 Report
The topic of water filled bodies is interesting and important for readers and engineers.
Below are the detailed comments.
- The abstract was organized in a bad logic.
- There are a lot of grammatical errors in the introduction and other parts. For example, Page 1, line 11, Page 2, line 43, Page 2, line 78, Page 4, line 126, Page 5, line 141, Page 8, line 226, Page 13, line 323.
- The experiment was conducted in a tank, but the informationabout the tank should be
- Cameras were used to capture the motion of the six markers and the motion of the rigid body calculated based on the motion of the markers. More details should be introduced in the paper.
- It is better to give a sketch of experimental set-up in the tank, including the coordinate, the location of the wave gauges, the floating cylinder and the wave direction, etc.
- The experimental conditions should be listed in a table, for example, pre-tensions, wave tank deployment locations and wave directions, etc.
- Would you like to elaborate a bit more discussions over the influence of the wave direction on the frequency band with a roll response? Why a change of the wave direction to 180°results in a significant increase? This is an important conclusion of the paper, more discussions are better for reader
Author Response

(The authors gave the same response as above.)

Reviewer 3 Report
Investigating the dynamic response of offshore structures and vessels under wave forcing is a complex issue relevant to the field of applied ocean and offshore engineering. The analysis requires suitable investigation approaches and modelling techniques [1-4]. In particular, the dynamic analysis of floating fluid-filled structures introduces additional complexity because the response of the body is affected by the interaction with the stored fluid undergoing large displacements and, possibly, breaking [5]. Some numerical models, based on the Smoothed Particle Hydrodynamics (SPH) method have been successfully developed for the analysis of such complex interactions [6-7]. Despite numerical modelling allows obtaining good results accuracy in reproducing laboratory experiments, physical modelling is complementary for model validation.
The present manuscript describes the results of laboratory experiments that extend a previous work on a fluid-filled floating cylinder subjected to roll and pitch motion. In particular, the present work focuses on the switch between pitch and roll in the cylinder under wave action as induced by the sloshing of the internal water volume. The influence of mooring system (i.e. pre-tension) and of the experiment duration is also investigated.
In my opinion, the manuscript can be considered for publication after major revision.
General Comments
The paper is clearly written.
English spelling is appropriate. Moderate English changes are required (see below).
The introduction section and the review of technical literature are quite suitable. Anyway, I suggest enhancing it by expanding a bit the topic of numerical modelling. As a help, some useful references are provided below.
Major concerns
As said by the Authors, the provided dataset includes a wide range of experiments with a clear pitch or roll response; this dataset can be used for the validation of numerical simulations.
Based on this statement, the main concern is the following.
The structural dynamics is coupled with the motion of the filling fluid that, however, has not been investigated in this study. What is the actual usefulness of these experimental data for validation of numerical models aiming at simulating the Fluid-Structure-Fluid interaction described in this work? Please carefully clarify this aspect.
Minor concerns
Line 82: the following sentence seems incorrect "The initial investigated compared the two ballast options"; may be "investigations" must replace "investigated"
Line 107: please change to "as a validation"
Line 140: please change to "with regular waves"
Lines 379, 394 and 409: is the use of bold type in the main text allowed by the journal standard?
References
- C. T. Stansberg Wave front steepness and influence on horizontal 3 deck impact loads. . Mar. Sci. Eng. 2020, 8(5), 314; https://doi.org/10.3390/jmse8050314
- Petrini F., Manenti S., Gkoumas K., Bontempi F. Structural design and analysis of offshore wind turbines from a system point of view. Wind Engineering Volume 34, Issue 1, 1 January 2010, Pages 85-108
- Manenti S., Leuzzi G., Monti P., Cerquarelli V. Wind-wave hindcasting on offshore wind turbine through coupled atmospheric and spectral models. Proceedings of the 12th International Conference on Engineering, Science, Construction, and Operations in Challenging Environments - Earth and Space 2010, Pages 2143-2151; Honolulu, HI; United States; 14 March 2010 through 17 March 2010; Code 81209.
- Manenti S., Petrini F. Dynamic analysis of an offshore wind turbine: Wind-waves nonlinear interaction. Proceedings of the 12th International Conference on Engineering, Science, Construction, and Operations in Challenging Environments - Earth and Space 2010, Pages 2014-2026; Honolulu, HI; United States; 14 March 2010 through 17 March 2010; Code 81209
- S. Huang, W. Duan, X. Han, R. Nicoll, et al. Nonlinear analysis of sloshing and floating body coupled motion in the time-domain. Ocean EngineeringVolume 16415 September 2018Pages 350-366
- Amicarelli A., S. Manenti, R. Albano, G. Agate, M. Paggi, L. Longoni, D. Mirauda, L. Ziane, G. Viccione, S. Todeschini, A. Sole, L.M. Baldini, D. Brambilla, M. Papini, M.C. Khellaf, B. Tagliafierro, L. Sarno, G. Pirovano. 2020 "SPHERA v.9.0.0: a Computational Fluid Dynamics research code, based on the Smoothed Particle Hydrodynamics mesh-less method". Computer Physics Communications, 250:107157; https://doi.org/10.1016/j.cpc.2020.107157.
- L. Delorme, A. Colagrossi, A. Souto-Iglesias, R. Zamora-Rodríguez, E. Botía-Vera. A set of canonical problems in sloshing, Part I: Pressure field in forced roll—comparison between experimental results and SPH Ocean Engineering Volume 36, Issue 2, February 2009, Pages 168-178
Author Response

(The authors gave the same response as above.)

Round 2
Reviewer 1 Report
The reviewers thank the authors for addressing the comments. The manuscript shall be recommended to be accepted at its present form.
Reviewer 2 Report
the authors have revised the manuscipt acoording to the reviewer's comments. the quality has been improved. it is reccomended to be published at its present stage.
Reviewer 3 Report
The manuscript is now suitable for publication.
This manuscript is a resubmission of an earlier submission. The following is a list of the peer review reports and author responses from that submission.
Round 1
Reviewer 1 Report
comments are all included in the marked up PDF

Author Response
We provide a detailed response in a separate document for all three reviewers. Thank you.

Reviewer 2 Report
1, what's the distance of the mooring line away from the water surface? How does it affect the results?
2, in general, I find the discussion of the results is not easy to follow. The authors may want to highlight the key results at the beginning or end of each section.
3, English writing should be improved.
Author Response

(The authors gave the same response as above.)

Reviewer 3 Report
The manuscript presents a series of experiment in a directional wave tank to assess the response of a floating moored structure ballasted with different loads.
There are numerous overlap with previous investigations by the same author, therefore is hard to identify the scientific novelty. This contribution appears as a simple extension of the database.
Presentation is poor. The plots do not facilitate the understanding of the experiments, often the authors plot on the x axis some combination of the variable plotted on the y axis. It is hard to clearly identify dependences, resonances, response. The plots often have label and marks too small.
Particularly puzzling is the analysis/discussion of long time series. Most of the plot clearly show that the system (wave + structure) produce some sort of interaction that establishes a coupled dynamical system (as shown by the variation in the amplitude of the rsponse), but the author derives response by using a single RAO. A different type analysis should be conducted, to investigate the response in the phase space.
I believe the manuscript not suitable for publication as it is. The analysis is poorly conducted, however the dataset might be valuable.
Author Response

(The authors gave the same response as above.)

Round 2
Reviewer 3 Report
Despite the attempts to justify the publication of the manuscript I do believe that the paper can not stand on its own, and I see as a trivial extension of a previous database. The database, very helpful indeed, must be published but doesn't constitute scientific novelty. For this reason I still believe the paper should be rejected.
I should add that the added figures where two degree of motion are shown are not phase diagram. They show however, that not even in the initial state a steady state is achieved, but the dynamical system oscillates, possibly between a couple of manifolds.
Labels and lines in plots are still small, and hardly legible.